# Early Life Adversity and Disordered Eating: Cognitive and Neural Mechanisms

**DOI:** 10.3390/bs15121739

**Published:** 2025-12-16

**Authors:** Yijun Luo, Jingqiu Zhang, Hong Chen

**Affiliations:** 1Key Laboratory of Cognition and Personality, Ministry of Education, Southwest University, Chongqing 400715, China; luoyijun@swu.edu.cn (Y.L.); zhangjingqiu2002@163.com (J.Z.); 2School of Psychology, Southwest University, Chongqing 400715, China

**Keywords:** early life adversity, disordered eating, neural mechanism, inferior frontal gyrus, structure–function coupling

## Abstract

The mosaic brain evolution perspective states that the relative sizes and functions of brain regions adapt to living environments and behavioural motivation. Early life adversity brings changes to brain structure, function, and patterns of cognitive processing of food cues. Specific brain development patterns are associated with subsequent disordered eating, which, on the one hand, increases the risk of obesity and metabolic syndrome, and, on the other hand, leads to mental health problems, such as depression and anxiety. This review intends to synthesise aberrant brain development indices, describe aberrant brain developmental trajectories, summarise aberrant neural markers of cognitive processing of food cues, conclude how early life adversity affects disordered eating through aberrant brain development patterns, and provide neural implications for future disordered eating research and intervention.

## 1. Introduction

On 27 April 2022, the General Office of the State Council of China issued the 14th Five-Year Plan for National Health, emphasising the importance of comprehensive interventions targeting health issues and their determinants. Among these, physical fitness has emerged as a national priority. Disordered eating refers to a broad range of maladaptive, psychologically driven eating behaviours that significantly impair physical health and psychosocial functioning ([9]). These behaviours elevate the risk of developing eating disorders, obesity, and metabolic diseases. For instance, even short-term binge eating can trigger inflammatory signalling at the cellular level, increase neutrophil counts, and accelerate the development of atherosclerotic plaques ([32]). In addition, our team found that chronic disordered eating is associated with increased symptoms of binge-eating disorder, bulimia nervosa, and higher obesity risk ([9]; [28]; [38]). Disordered eating has also been linked to psychological problems such as depression, anxiety, self-denial, and even self-injurious behaviours ([71]). Therefore, investigating the risk factors for disordered eating and its underlying cognitive-neural mechanisms is crucial for the development of effective prevention and intervention strategies for related conditions, such as depression, anxiety, eating disorders, and obesity. Such research holds direct practical relevance for efforts to improve public health and also contributes to the global understanding of eating disorders.

Since 2008, our research team has systematically investigated the psychological and behavioural factors contributing to disordered eating. We have identified sociocultural pressure, negative body image, attentional and memory biases toward food-related cues, and impaired conflict monitoring as significant risk factors for maladaptive eating behaviours ([35]; [41], [39]). More recently, large-scale longitudinal studies, behavioural experiments, and neuroimaging techniques, have collected initial evidence suggesting that early life adversity (ELA) may increase the risk of disordered eating in children and adolescents through changes in life history strategies and functional connectivity of the amygdala ([37], [42], [38], [40]).

ELA refers to adverse experiences occurring between birth and age 16, either within the family or broader social environment. These experiences include physical, emotional, and sexual abuse as well as neglect, economic hardship, parental separation or divorce, and exposure to domestic violence ([44]). Despite increased societal awareness of ELA, scholars emphasise its prevalence as a public health challenge and a robust predictor of poor mental and physical health outcomes ([63]). Similarly, a national policy issued by the General Office of the State Council of China in December 2024 called for improved early development services, highlighting the urgent need to prevent and mitigate the long-term health consequences of ELA. Yet current interventions are primarily behavioural or pharmacological, with limited targeting of the underlying neural mechanisms. This study aimed to review the cognitive-neural pathways linking ELA and disordered eating to provide more effective, mechanism-based prevention and intervention strategies.

### Neural Development Patterns of ELA

Mosaic brain evolution perspective suggests that different brain regions may experience adaptive changes in their relative sizes in response to environmental pressures and behavioural demands. Such region-specific neural developmental patterns are thought to reflect the selective evolution of distinct sensory and cognitive capacities ([15]). Hence, this evolutionary framework provides a compelling lens through which to interpret the neural consequences of ELA. From an adaptive perspective, ELA may signal an unstable or threatening environment in which long-term investments yield limited survival benefits. As a result, individuals experiencing ELA may adopt an accelerated life history strategy that prioritises immediate rewards. Structural and functional changes in brain regions responsible for reward sensitivity and inhibitory control may be a maladaptive consequence of this adaptive calibration ([18]).

Consistent with this view, a growing body of research has documented region-specific alterations in brain development following ELA. For example, emotion- and reward-related areas (e.g., the amygdala, and nucleus accumbens) tend to show accelerated maturation, whereas prefrontal regions implicated in inhibitory control often exhibit delayed development ([8]; [27]; [51]; [70]). These findings suggest that ELA may trigger a mosaic pattern of brain development with differential maturation timing across functional systems, reflecting an adaptive response to adversity. However, this adaptive response may compromise long-term emotional regulation, impulse control, and reward processing.

## 2. Method

### Paper Identification and Review

The literature search followed PRISMA guidelines. The full screening workflow is presented in Figure 1. To identify papers to include in the systematic review, we searched PubMed database and Nature Affiliated journals for studies conducted in humans on childhood adversity and disordered eating between 1 January 2015 and 1 May 2025. Childhood adversity was indexed using the following search phrases: “early life stress”, “childhood adversity”, “childhood trauma”, “adverse childhood experiences”, “childhood maltreatment”, “emotional abuse”, “emotional neglect”, “physical abuse”, “sexual abuse”, “physical neglect”, “poverty”, and “environmental unpredictability”. Cognitive and neural mechanism was indexed using “cognitive”, “neural”. Disordered eating was indexed using “overeating”, “emotional eating”, “disordered eating”, “uncontrolled eating”, “binge eating”.

We included studies that: (1) examined early life adversity (ELA), or subclinical disordered eating symptoms/disordered eating; (2) used neuroimaging methods (structural MRI, resting-state fMRI, or task-based fMRI). We excluded studies that: (1) did not involve neuroimaging, such as genetic research or behavioural research; or (2) focused on clinically diagnosed populations, including eating disorders or other psychiatric or medical conditions.

The initial search yielded 4978 records (PubMed: 3514; Nature and affiliated journals: 1464). After title and abstract screening, 3924 records were removed for being clearly unrelated to the topic. An additional 571 news items, commentaries, or narrative reviews were excluded. We then removed 3353 records that did not involve human neuroimaging. A total of 1054 articles underwent full-text assessment, during which 966 were excluded because they involved clinical populations or did not meet other inclusion criteria. After removing 41 duplicates, 47 studies met all criteria and were included in the final analysis.

Among the 47 studies included, the age composition of samples differed substantially. Based on the mean age reported in each study, 7 examined early childhood groups (0–6 years), 20 involved school-age or adolescent samples, and 20 focused on adult participants. Notably, because early-life adversity is generally defined as exposure occurring before age 16.

## 3. Results

### 3.1. Structural Changes, Functional Changes, and Cognitive Processing Patterns Related to ELA

In neuroimaging studies reviewed in latest 5 years (Table 1), structural changes related to ELA showed a certain pattern. For example, [27] ([27]) conducted a large-scale cohort study employing machine learning techniques and voxel-based normative modelling to quantify the impact of ELA on brain morphology at the structural level. Their findings revealed widespread structural alterations associated with ELA, including increased volume of the amygdala, hippocampus, and orbitofrontal cortex (OFC), as well as reduced volume of the ventromedial prefrontal cortex (vmPFC) and anterior cingulate cortex (ACC). Importantly, volumetric alterations of the OFC persisted into adulthood, remaining stable at both 25 and 33 years. These results indicate that ELA may accelerate the maturation of emotion- and reward-related regions while delaying the development of brain areas involved in inhibitory control. Such a region-specific developmental trajectory lends empirical support to the ‘accelerated–delayed’ brain development hypothesis in response to early environmental stressors.

Functional changes associated with ELA focused on emotional regulation relevant region and network. ELA may lead to premature maturation of emotion perception systems, while concurrently disrupting the development of emotion regulation capacities. This is evidenced by increased negative functional connectivity between the prefrontal cortex and the amygdala ([25]). In addition, the intrinsic variability in brain networks—such as time series variability and functional coupling variability—has been proposed as a marker of the brain’s adaptive capacity to environmental demands. Time series variability reflects a region’s sensitivity to external changes, whereas coupling variability captures the stability of functional integration across networks. In a sample of adolescent girls, increased functional coupling variability was associated with accelerated neurodevelopment, particularly in the visual network, attentional network, and default mode network (DMN). These findings suggest that, in addition to sensory-related systems, variability in networks subserving higher-order cognitive, affective, and social processing (i.e., the DMN) may be a signature of accelerated maturation ([51]). Moreover, structure–function coupling (SC–FC) is considered to be a key indicator of brain maturation and neural plasticity. Between the ages of 8 and 22 years, SC–FC changes occur in a functional-network-specific manner, with decreases in highly conserved motor regions and increases in transmodal cortices ([2]). A recent birth cohort study tracking 549 children between ages 4.5 and 7.5 years revealed a linear decline in whole-brain structure–function coupling, suggesting a normative age-related reduction in coupling strength. Notably, this decline was steeper in children exposed to high levels of ELA, and this is indicative of accelerated brain development. This acceleration was most evident in transmodal integrative networks, such as the frontoparietal network; however, it was not observed in unimodal sensory networks ([8]). These findings suggest that ELA may expedite the maturation of higher-order brain systems as an adaptive response to environmental challenges. Furthermore, this maturation is potentially at the cost of reduced neural plasticity required for later functional refinement.

Heightened sensitivity in reward-related circuits and the delayed development of executive functions are considered to be key neural markers of atypical brain development associated with ELA. [25] ([25]) reported that children exposed to early adversity exhibit increased activation in the nucleus accumbens and medial prefrontal cortex during reward tasks, suggesting heightened reward responsivity. Similarly, in gambling paradigms, individuals with histories of ELA display faster and more impulsive decision-making. They opt for high-risk, high-reward choices despite repeated losses, and fail to adjust their strategies to minimise negative outcomes. These behaviours are accompanied by aberrant activation in reward-related regions including the putamen, insula, and precuneus ([3]). Furthermore, children who have experienced high adversity levels demonstrate poorer executive function than their low-adversity peers by age eight years on measures of attention, short-term visual memory, and spatial working memory. Moreover, the developmental trajectories of these functions slow over time, contributing to a widening gap in cognitive performance by adolescence ([64]).

In summary, individuals exposed to ELA may exhibit accelerated development in neural circuits involved in reward and emotion processing. This is thought to be an adaptive response to a threatening environment. However, this adaptation may come at the cost of delayed maturation in brain regions supporting inhibitory control, potentially compromising regulatory capacity in later stages of development. These neurodevelopmental changes may contribute to maladaptive eating behaviours, particularly those characterised by heightened reward sensitivity and diminished impulse control ([71]).

### 3.2. Association Between ELA and Disordered Eating

Disordered eating behaviours involve the interplay of three key neural systems: the emotional processing system, the reward circuitry, and the inhibitory control network. Emerging evidence indicates that ELA may accelerate the development of emotion- and reward-related brain regions, while simultaneously delaying the maturation of areas responsible for inhibitory control ([3]; [64]; [10]). This imbalance may compromise individuals’ capacity to regulate responses to food-related rewards, thereby increasing the likelihood of maladaptive eating behaviours. Prior research on the neurocognitive mechanisms of disordered eating has primarily focused on the interaction between reward sensitivity and inhibitory control; however, it is essential to consider the regulatory role of the emotional processing system in the ELA context. Structural and functional alterations in emotion-related circuits may modulate both reward responsivity and impulsivity, thereby indirectly contributing to the onset and maintenance of disordered eating.

#### 3.2.1. Structural MRI Evidence

ELA may influence disordered eating behaviours through structural alterations in the brain, particularly in regions involved in emotion regulation and cognitive control ([20]; [22]). Neuroimaging studies have consistently associated early adversity with reduced prefrontal cortical volume and impaired executive functions, including goal-directed behaviour, working memory, and emotional regulation ([34]). In contrast, emotion-related circuits such as the amygdala show heightened reactivity and enhanced emotional memory, reflecting a developmental imbalance between affective and regulatory systems.

The inferior frontal gyrus (IFG), a key region within the prefrontal cortex, appears to be particularly affected. Adolescents who engage in emotional or uncontrolled eating often display delayed maturation of the prefrontal cortex, including the IFG and cerebellum ([70]; [5]). Longitudinal studies have linked ELA to reductions in overall brain volume, including an 8.6% decrease in total brain volume and significant reductions in the surface area and volume of the right IFG.

White matter abnormalities further support this pattern. Children exposed to ELA exhibit decreased mean diffusivity and increased fractional anisotropy in key tracts connecting the prefrontal cortex and limbic system. These tracts include the cingulum bundle, uncinate fasciculus, and fornix. Reductions in cortical thickness of the IFG have also been observed in maltreated adolescents ([22]). Additionally, IFG abnormalities are consistently reported in populations with ADHD, bipolar disorder, and eating disorders ([20]).

In summary, ELA may alter structural development of the IFG, compromising inhibitory control and increasing susceptibility to reward-driven disordered eating.

#### 3.2.2. Resting States and Task-Based fMRI Evidence

ELA may increase the risk of disordered eating by altering the neural architectures of emotion regulation, reward processing, and inhibitory control. Resting-state fMRI studies have associated self-reported childhood adversity with spontaneous activity in the basolateral and centromedial subregions of the amygdala. In particular, adversity scores can predict both functional connectivity between the bilateral basolateral amygdala and the left IFG and disordered eating behaviour one year later ([37]). In another retrospective study, adversity scores were positively correlated with attentional bias toward food cues and negatively associated with functional connectivity between the right IFG and the left inferior parietal lobule. This connectivity, in turn, predicted disordered eating severity ([42]). These findings suggest that early adversity may impair prefrontal-parietal circuits underlying inhibitory control, thereby contributing to increased food-related impulsivity.

Task-based fMRI studies further provide evidence for these associations. Compared to women with low levels of ELA, those who had experienced high adversity displayed heightened activation in the amygdala, dorsal striatum (caudate and putamen), medial OFC, and ACC during food-related tasks. Moreover, increased amygdala–putamen connectivity and decreased amygdala–ACC/PFC connectivity were observed in the high-adversity group ([62]). This suggests a shift toward stronger emotion-to-reward coupling and weaker emotion-to-control integration among individuals with high levels of ELA. Findings from animal models align with these human data. In rats exposed to ELA and later treated with opioid agents, there was reduced activation in the nucleus accumbens core, whereas the central amygdala and prefrontal cortex showed increased activation ([33]). This suggests that early adversity may reshape neural activity within emotion, reward, and inhibitory control circuits, thereby altering sensitivity to food-related stimuli.

#### 3.2.3. Cognitive Processing Patterns

ELA may increase the risk of disordered eating by altering how individuals process food-related reward cues. These alterations are observed across three stages of reward processing—anticipation, consumption, and decision-making. They involve brain regions implicated in emotion, reward, and cognitive control.

During the anticipation of food rewards, individuals with high levels of ELA exhibit heightened activation in brain regions related to gustatory processing (e.g., insular cortex), somatosensory integration, and reward valuation (e.g., the amygdala, and vmPFC) ([26]). This increased activation may reflect stronger motivation to seek food to alleviate persistent negative affect stemming from early-life stress.

In contrast, when receiving food rewards, these individuals show attenuated activation in key reward-related regions, such as the dorsal striatum (e.g., caudate nucleus) ([58]). This discrepancy between high anticipatory and low consummatory responses can be explained using two theoretical frameworks. The first is incentive sensitization theory, which posits an increased sensitivity to reward cues but a diminished response to the reward itself. The second is the reward deficiency hypothesis, which suggests that food rewards fail to fully compensate for the emotional deficits caused by ELA, resulting in a chronic reward prediction error.

During the decision-making phase, ELA appears to compromise self-regulation. Higher levels of adversity are associated with stronger activation in the vmPFC, amygdala, and striatal regions when evaluating palatable food options. These areas are known to encode hedonic value, and their heightened activity predicts the selection of high-calorie over low-calorie foods ([43]). Furthermore, ELA disrupts effective connectivity among the prefrontal cortex, limbic system, and striatum, weakening the neural mechanisms underlying inhibitory control. Individuals with high ELA may exhibit a maladaptive reward processing profile characterised by heightened anticipation, blunted reward experience, and impulsive food choices. These abnormalities involve both classic reward-related areas (e.g., the amygdala, striatum, OFC, ACC) and regions responsible for higher-order control (e.g., dorsolateral prefrontal cortex). This is a neural basis for increased susceptibility to disordered eating.

Considering the wide-ranging and long-lasting effects of disordered eating, a deeper understanding of how ELA associated brain changes contribute to disordered eating may identify novel brain circuit targets for advanced treatments and preventive strategies.

## 4. Discussion

As a systematic review, this study synthesises evidence across multiple domains to examine the cognitive and neural mechanisms linking Early Life Adversity (ELA) to subsequent disordered eating, highlighting specific patterns of brain development that may underlie this association. Results indicate that ELA exerts pervasive effects on the development and function of neural circuits. Specifically, it appears to accelerate the maturation of emotion- and reward-related brain regions while delaying the development of areas responsible for inhibitory control.

Therefore, efforts to promote children’s physical and mental development should prioritise reducing adverse factors embedded in the family, school, and community environments. Additionally, future research should investigate how the two distinct dimensions of ELA—specifically, abuse and neglect—differentially shape this neural development trajectory.

Despite the identified imbalance (accelerated emotion/reward versus delayed inhibitory control), few studies have directly related this pattern to subsequent disordered eating symptoms. Moving forward, integrating findings from clinical and preclinical research will be essential for identifying the precise neurobiological mechanisms that causally mediate the effects of ELA on maladaptive eating. This neuro-developmental imbalance may fundamentally impair an individual’s capacity to regulate responses to food-related rewards, thereby heightening the likelihood of maladaptive eating behaviours.

Understanding these neural mechanisms not only helps to eliminate the stigma associated with abused or neglected children—as their behaviours are manifestations of a brain injury, not them being “innately bad”—but also provides precise targets for scientific intervention.

The most crucial implication of these findings is the recognition that these ELA-related brain changes are often adaptive responses rather than permanent damage, underscoring the remarkable neuroplasticity available during childhood and adolescence. Interventions which provide a stable, safe, responsive, and supportive environment—such as high-quality alternative care, trauma-focused psychotherapy, and school-based support—can actively help these children’s brain development trend toward repair and recovery.

## 5. Limitations

However, several limitations of this review should be clarified. First, we did not examine potential moderators of the neurodevelopmental pathways linked to disordered eating. Our focus was on how ELA-related patterns—accelerated maturation of emotion–reward systems and delayed development of inhibitory control—may converge to shape vulnerability. As a result, other influences such as gender, age, socioeconomic status, or comorbid conditions were not considered. These factors remain important, and future work should test how they modify the link between early adversity and eating-related outcomes at different developmental stages.

Second, the imbalance model outlined here is conceptual and derived from indirect evidence. Its assumptions have not been tested through direct statistical modelling, and any causal interpretation should be viewed as provisional. Longitudinal neuroimaging studies will be essential for validating or revising this framework.

Finally, most studies included in this review focused on early childhood, adolescence, and young adulthood. This concentration in younger samples should be noted, as it limits the generalisability of the findings to later developmental periods.

## 6. Conclusions

Early life adversity (ELA) exerts pervasive effects on the development and function of neural circuits, thereby increasing the risk of disordered eating and obesity across the lifespan. Specifically, it can accelerate the maturation of emotion- and reward-related brain regions while delaying inhibitory control related regions. This neurodevelopmental imbalance may impair an individual’s capacity to regulate responses to food-related rewards, thereby heightening the likelihood of maladaptive eating behaviours and increasing the risk of obesity. Addressing how ELA disrupts the intricate interplay among the emotion, reward, and inhibitory control systems remains a major challenge for contemporary neuroscience and psychiatry, requiring consideration of both the molecular regulation of these systems and the developmental trajectories of individual brain regions.

## Figures and Tables

**Figure 1 behavsci-15-01739-f001:**
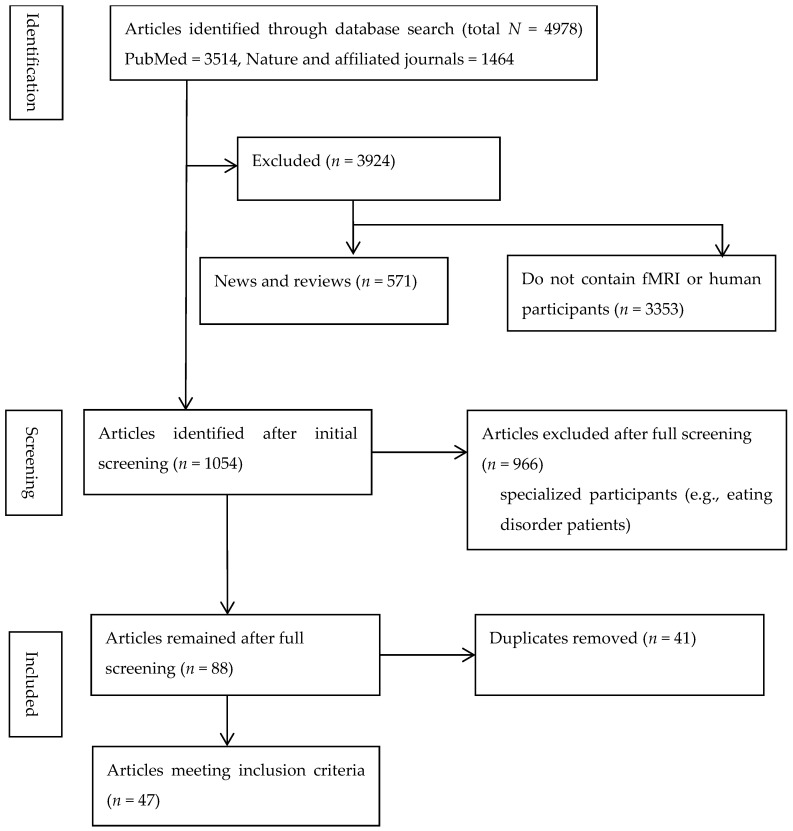
PRISMA flow diagram detailing the literature search and selection process for inclusion in the systematic review.

**Table 1 behavsci-15-01739-t001:** Summary for Neuroimaging Studies on Early Life/Childhood Experiences and disordered eating.

Study	Sample Size	Age ^1^ (M ± SD)	Measures of Adversity	Neuroimaging	Task	Main Findings
**Summary for Neuroimaging Studies on Early Life/Childhood Experiences**						
([6])	1184	9.54 ± 0.86	Adverse childhood experiences	MRI		Substance abuse in the household was associated with larger cortical surface area in the left superior frontal gyrus, right superior frontal gyrus, left pars triangularis, left rostral middle frontal gyrus, and right caudal anterior cingulate gyrus. Household exposure to violence was associated with lower fractional anisotropy in the left and right cingulum bundle hippocampus region.
([7])	155	11.33 ± 1.05	The Traumatic Events Screening Inventory for Children; modified version of the UCLA Life Stress Interview	MRI		Severity of ELA was associated with greater likelihood of belonging to the low HCV/high symptom
([8])	549	31.1 ± 5.2	Adversity measured by seven components focused on prenatal exposures	MRI, RSFC		A linear decrease in SC–FC was observed from age 4.5 to 7.5 years. When stratified by ELA, only the high-adversity group showed a curvilinear trajectory, with a steep decrease between age 4.5 and 6 years, suggestive of accelerated neurodevelopment.
([14])	Adversity, *n* = 127; None, *n* = 96; Abuse only, *n* = 33; Neglect only, *n* = 28; Both, *n* = 45; Neither, *n* = 96	16.1 ± 2.3; 16.4 ± 2.5; 16.7 ± 2.5; 15.5 ± 2.3; 16.4 ± 2.3; 16.4 ± 2.5	Adverse Childhood Experiences (ACE) questionnaire; Childhood Trauma Questionnaire	RSFC		In a general model, adversity was associated with altered amygdala RSFC with clusters within the left anterior lateral prefrontal cortex. In a dimensional model, abuse was associated with altered amygdala rs-fc within the orbitofrontal cortex, dorsal praecuneus, posterior cingulate cortex, and dorsal anterior cingulate cortex/anterior mid-cingulate cortex, as well as within the dorsal attention, visual, and somatomotor networks. Neglect was associated with altered amygdala RSFC with the hippocampus, supplementary motor cortex, temporoparietal junction, and regions within the dorsal attention network
([16])	85	Maternal age at delivery 31.37 ± 5.17	Adverse Childhood Experiences Questionnaire	MRI		Higher maternal ACEs were associated with smaller left, and right amygdala volume of infants.
([17])	91	16.46 ± 0.52	Event planning and problem-solving interactions measuring mother-adolescent dyads; stressful life events questionnaire and the Child Trauma Questionnaire	MRI		In females, childhood maltreatment predicted larger anterior pituitary gland volume at T4.
				RSFC		ELA were associated with decreased anterior hippocampal-cortical functional connectivity.
([19])	338	9.9 ± 0.1	Adverse Life Events Scale	RSFC		Adverse life events were related to greater decreases in network-to-subcortical functional connectivity across most networks, but particularly for the cingulo-opercular network and sensorimotor network; adverse life events were also associated with increasing connectivity within and between motor and sensory networks.
([24])	48	26.4 ± 2.7 (weeks)	Childhood Trauma Questionnaire–Short Form; Perceived Stress Scale; Spielberger State-Trait Anxiety Inventory; Edinburgh Depression Scale	RSFC		Emotional neglect from the mother’s childhood positively correlated with left and right amygdala–the dorsal anterior cingulate cortex functional connectivity in neonates
([27])	169	29.60 ± 5.92	Childhood Trauma Questionnaire	MRI		Structural alterations were associated with ELA, including increased volume of the amygdala, hippocampus, and orbitofrontal cortex (OFC), as well as reduced volume of the ventromedial prefrontal cortex (vmPFC) and anterior cingulate cortex (ACC)
([29])	9270	119.17 ± 7.47 (months)	Kiddie Schedule for Affective Disorders and Schizophrenia (K-SADS) administered to a parent or guardian	MRI		Trauma exposure was associated with thinner cortices in the bilateral superior frontal gyri and right caudal middle frontal gyrus as well as thicker cortices in the left isthmus cingulate and posterior cingulate; trauma exposure was associated with smaller GMV in the right amygdala and right putamen.
([30])	400	40.6 ± 10.4	short form of Egna Minnen Beträffande Uppfostran; Childhood Trauma Questionnaire	MRI		Greater parental rejection resulted in smaller hippocampal and amygdala volumes;
([31])	77	181 ± 15 (months)	Youth Life Stress Interview	MRI		Greater adversity was associated with lower accumbofrontal tract integrity in both the left and right hemispheres
([37])	85	10.20 ± 0.99	Family Unpredictability Scale; subjective SES; Children’s Eating Behaviour Questionnaire; Food portion choice in the absence of hunger	RSFC		Environmental harshness and unpredictability were negatively associated with bilateral BLA-left inferior frontal gyrus (IFG) connectivity, while dynamic RSFC analyses found that environmental harshness and unpredictability was negatively associated with right CMA, left inferior parietal lobule, and right CMA-right precuneus connectivity
([42])	501	19.22 ± 0.83	Environmental Harshness and Unpredictability Scale; uncontrolled eating subscale of eating behaviours	RSFC		Harsh, unpredictable childhood environments are associated with significant but modest decreases in connectivity of right inferior frontal gyrus (IFG)-bilateral medial frontal gyrus, right IFG-bilateral inferior parietal lobule (IPL), and right IFG-left superior frontal gyrus connectivity, as well as attentional engagement to high-calorie food and binge eating tendencies.
([47])	275	21.53 ± 1.46	cumulative psychological trauma; cumulative pre and perinatal risk	MRI		Cumulative pre/perinatal risk was associated with smaller left subgenual cingulate volume; Cumulative childhood trauma was associated with larger left dorsal striatum, right prefrontal cortex and smaller left insula volume
([48])	115	11.51 ± 1.08	modified version of the Traumatic Events Screening Inventory for Children; modified version of the UCLA Life Stress Interview coding system	MRI		ELA was positively associated with change in brain volume (expansion) in 13 of these clusters, which included white matter in the left cerebellum and bilateral superior frontal gyrus, and grey matter in the left caudate, right cuneus, right lateral and inferior occipital gyrus, and various frontal cortical regions (e.g., bilateral superior and inferior frontal gyrus, bilateral medial orbitofrontal cortex). ELA was negatively associated with volumetric change (contraction) in 9 clusters that included gray matter in the right globus pallidus, right entorhinal cortex and fusiform gyrus, left posterior cingulate gyrus, and various frontal, temporal, and parietal cortical regions (e.g., right superior frontal gyrus, left supramarginal gyrus)
([49])	784	At age 4.5, 6, 7.5	Perinatal adversity	MRI		Greater adversity was associated with reduced bilateral hippocampal body volume in early childhood, but also to faster growth in the right hippocampal body across childhood.
([50])	46	4.8 ± 0.8	Early life stress, Exposure to parental depression, Exposure to parental hostility	MRI & fMRI & RSFC	Reward Processing Task	In the reward condition, higher levels of early life stress, were related to decreased right amygdala connectivity with frontal regions during hit trials but increased connectivity during miss trials. Higher levels of early life stress were associated with greater differences in left amygdala connectivity with left dorsal frontal cortex
([52])	161	12.64 ± 2.67	Childhood Experiences of Care and Abuse Interview; Childhood Trauma Questionnaire; Violence Exposure Scale for Children Revised; short form of the Home Observation for Measurement of the Environment	MRI		Greater threat was associated with thinner cortex in a network including areas involved in salience processing (anterior insula, vmPFC), and smaller amygdala volume (particularly in younger participants), after controlling for deprivation. Threat was also associated with thinning in the frontoparietal control network.
([53])	Maltreatment, *n* = 96; Comparison, *n* = 288	19.20 ± 0.79; 19.04 ± 0.71	Childhood Trauma Questionnaire	MRI		Individuals with a maltreatment history had reduced surface areas and cortical thicknesses primarily in frontal-temporal regions; and they also had larger cortical thicknesses in occipital regions and surface areas in frontal regions.
([54])	381	7.5 cohort study	State–Trait Anxiety Inventory; Beck Depression Inventory—II, the Edinburgh Postnatal Depression Scale; questions on pregnancy-related feelings administered at 26–28 weeks of Maternal pregnancy	RSFC, MRI		Girls born to mothers who reported greater positive mental health during pregnancy showed larger bilateral hippocampi.; Children of mothers with greater positive mental health exhibited altered functional connectivity of several networks, including default mode, salience, executive control, amygdala and thalamo-hippocampal networks
([55])	160	40.08 ± 13.64	Childhood Adversity Questionnaire	MRI		Trauma exposure was negatively associated with GMC of the middle frontal gyrus and parietal lobule, while negatively associated with grey matter volume covariation in the cerebellum.
([57])	161	32.21 ± 0.29	modified version of Munich Event List; block-designed emotion regulation task	RSFC		While prenatal and childhood stress were associated with lower connectivity between subcortex and cognitive networks, stress exposure unique to adolescence was related to higher connectivity from the salience network to the cognitive networks during emotion regulation.
([61])	261	41.3 ± 8.1 (months)	Area Deprivation Index percentiles; Healthy Eating Index; Prenatal disadvantage	RSFC		The magnitude of associations between prenatal disadvantage and developmental increases in local segregation differed across functional systems, with the strongest associations found in somatomotor-hand, somatomotor-mouth, dorsal attention, visual, and frontoparietal systems
([65])	158	22.07 ± 2.08	Childhood unpredictability	MRI		After controlling for the effect of childhood trauma, childhood unpredictability was correlated with greater GMV in bilateral frontal pole, bilateral precuneus, bilateral postcentral gyrus, right hemisphere of fusiform, and lingual gyrus, and left hemisphere of ventrolateral prefrontal cortex as well as occipital gyrus
([66])	Normative, *n* = 245; Low, *n* = 377; Moderate, *n* = 376; High, *n* = 376	14.3 ± 4.2; 13.8 ± 3.3; 14.9 ± 3.4; 16.8 ± 3.1	Traumatic Stressful Load; Stressor Reactivity Score	MRI		The high SRS group revealed a pattern of accelerated grey matter maturation; Positive correlations between averaged quantile regression index and Stressor reactivity score were found in bilateral thalamus proper and putamen, right caudate and amygdala
([68])	5885	119.13 ± 7.51 (months)	Life Events Scale	RSFC		Unpredictability was associated with a smaller increase in RSFC within default mode network (DMN) and a smaller decrease in RSFC between cingulo-opercular network (CON) and DMN.
([69])	5885	119.13 ± 7.51 (months)	Interpersonal and socioeconomic threat; Interpersonal and socioeconomic unpredictability; Interpersonal and socioeconomic deprivation	RSFC		Interpersonal unpredictability was associated with a greater decrease in RSFC of cingulo-opercular network(CON)-left amygdala, CON-right amygdala, CON-right hippocampus, and retrosplenial temporal network(RTN)-left hippocampus, as well as a greater increase in RSFC of RTN-right hippocampus; socioeconomic deprivation was associated with a greater decrease in RSFC of CON-left amygdala, CON-right amygdala, CON-right hippocampus, and RTN-left hippocampus, as well as a greater increase in RSFC of RTN-left amygdala and RTN-right hippocampus.
([72])	215	25.5 ± 6.3	Self-reported retrospective childhood trauma questionnaire	RSFC		FPN and DN networks made large contribution to predicting emotional abuse and physical neglect, FPN, DN, and VIS networks made large contribution to predicting emotional neglect, FPN, DAT, and VAT networks made large contribution to predicting sexual abuse, and VIS, DN, FPN, and VAT showed large contribution to predictingphysical abuse.
([73])	202	23.2 ± 1.7	Maltreatment and Abuse Chronology of Exposure scale	fMRI	Food Pictures Task	BOLD activation fMRI response to threatening versus neutral facial images was assessed in key components of the threat detection system (i.e., amygdala, hippocampus, anterior cingulate, inferior frontal gyrus and ventromedial and dorsomedial prefrontal cortices);
([21])	521	10.47 ± 2.79	Alabama Parenting Questionnaire (APQ), Barratt Simplified Measure of Social Status (Barratt), Family History—Research Diagnostic Criteria (PreInt_RDC), Financial Security Questionnaire (FSQ), Negative Life Events Scale (NLES) and PhenX neighbourhood Safety (PhenX)	Task-based fMRI	Viewing Emotion-specific and Emotion Non-specific Content	DMN, VAN, CON and amygdala activation during sad/emotional, bonding, action, conflict, sad, or fearful scenes; Greater inconsistent discipline was associated with greater VAN activation during sad or emotional scenes.
([67])	45	14.9 ± 1.9	Childhood Trauma Questionnaire	Task-based fMRI	Monetary Incentive Delay Task	Adolescents who experienced higher levels of abuse show greater activation in right dorsal prefrontal cortex, right inferior frontal gyrus, and right ventrolateral prefrontal cortex in the non-reward compared to reward condition during reward anticipation; individuals with higher abuse show much more exaggerated differences among conditions, especially during misses: greater activation when missing a potential reward vs. missing when there was no potential reward in multiple temporoparietal (left temporoparietal junction, right middle/superior temporal gyrus, right precuneus, right temporal pole, right Para hippocampal gyrus, left middle/inferior temporal and fusiform gyri, right angular/inferior parietal lobule, left precuneus/angular gyrus), posterior (left precuneus/posterior cingulate gyrus), and prefrontal (left dorsolateral) cortical regions during performance.
**Summary for Neuroimaging Studies on disordered eating**						
([4])	Young, *n* = 21; Old, *n* = 20	23.59 ± 4.22; 67.01 ± 3.68	Three Factor Eating Questionnaire	RSFC		Older, compared to younger, individuals reported lower levels of disinhibited eating, consumed a healthier diet, and had weaker connectivity in the frontoparietal (FPN) and default mode (DMN) networks.
([11])	693	18.37 ± 0.87	Eating disorder diagnosis scale	RSFC		Bulimia-type eating was associated with weaker intra-network and inter-network functional synchrony;
([13])	76	9.86 ± 0.83	The eating disorder inventory-child-bulimia subscale	RSFC, MRI		Higher levels of BE were correlated with greater grey matter volumes (GMV) in the left fusiform and right insula and weaker RSFC between the right insula and following three regions: right orbital frontal cortex, left cingulate gyrus, and left superior frontal gyrus
([10])	660	18.51 ± 1.04	Eating disorder diagnosis scale	RSFC		The connectivity predictive of body image concerns was identified within and between networks implicated in cognitive control (frontoparietal and medial frontal), reward sensitivity (subcortical), and visual perception (visual).
([36])	sub-BN Group, *n* = 145; Control Group, *n* = 140	18.99 ± 0.76; 18.89 ± 0.91	Eating disorder diagnosis scale	MRI; RSFC		The sub-BN group exhibited abnormalities of the right dorsolateral prefrontal cortex and right orbitofrontal cortex in both GMV and DC, and displayed decreased FC between these regions and the precuneus. We also observed that sub-BN presented with reduced FC between the calcarine and superior temporal gyrus, middle temporal gyrus and inferior parietal gyrus.
([45])	BED symptoms, *n* = 83; Control Group, *n* = 123	9.9 ± 0.60; 10.0 ± 0.60	Kiddie Schedule for Affective Disorders and Schizophrenia	RSFC		The BED-S group showed alterations in topological properties associated with the frontostriatal subnetwork, such as reduced nodal efficiency in the superior frontal gyrus, nucleus accumbens, putamen.
([46])	BE symptoms, *n* = 77; Control Group, *n* = 104	9.95 ± 0.62; 9.97 ± 0.60	Parent/guardian responses to the Kiddie Schedule for Affective Disorders and Schizophrenia (K-SADS) based on DSM-5 criteria; parent reports of binge eating (ABCD K-SADS item: Symptom—Binge Eating Present)	RSFC		The nodal topological properties, i.e., nodal efficiency, betweenness-centrality and degree, of the caudate nucleus, hippocampus and inferior parietal gyrus (IPG) significantly differentiated children with and without BE symptoms;
([56])	No food addiction, *n* = 72; Food Addiction, *n* = 42	33.24 ± 10.92 30.33 ± 9.35	Early Traumatic Inventory-Self Report; Yale Food Addiction Scale	MRI		Individuals with high FA had greater cortical thickness of the ACC, and greater surface area and volume of the orbital gyrus, while individuals with low FA had greater surface area and volume of the posterior central sulcus; participants with FA had greater volume of the caudate nucleus, greater cortical thickness of the short insular gyrus, and greater volume of the laterodorsal tegmentum nucleus; participants with the highest levels of ELA showed the strongest negative association between the Reward Control brain signature and FA scores.
([60])	212	18.87 ± 0.97	Muscularity-Oriented Eating Test; Childhood Trauma Questionnaire	RSFC		CPM identified the most significant predictive connections in the dorsolateral prefrontal cortex, inferior frontal gyrus, and cerebellum, with positive muscularity-oriented disordered eating networks primarily linking the salience/limbic network to the cerebellum and the fronto-parietal network to the default mode network.
([71])	REs, N = 324; E/UEs, N = 249; HEs, N = 423	14.56 ± 0.43; 14.49 ± 0.41; 14.49 ± 0.42	ED section (section P) of the Development and Well-being Assessment; short version of the TFEQ	MRI		Compared with HEs, REs showed smaller grey matter volume (GMV) reductions in the left cerebellum. E/UEs had smaller GMV reductions in two subclusters in the left cerebellum and five subclusters in the right hemisphere, including the middle frontal gyrus (MFG), putamen, medial superior frontal gyrus and postcentral gyrus. A lower GMV increase in the right parahippocampal gyrus was also observed; for cortical thickness (CT), with REs experiencing more pro nounced CT reduction. In contrast, comparing E/UEs with HEs showed significant interactions across nine brain regions. These included the left lingual gyrus, bilateral frontal pole, bilateral rostral MFG, left pericalcarine, left cuneus, right caudal MFG and right caudal anterior cingulate gyrus, with E/UEs showing less CT reduction than HEs; for sulcal depth (SD), REs had a less pronounced reduction in the left frontal pole compared with E/UEs. Moreover, E/UEs showed larger SD reductions across nine regions compared with HEs, including the bilateral rostral MFG, left frontal pole, bilateral superior frontal gyrus, right caudal MFG, right pars orbitalis, right pars opercularis and right pars triangularis gyrus
([1])	43	31 ± 7.8	—	Task-based fMRI	Food Pictures fMRI task	Regions positively associated with both Pleasantness and Self-Control included reward-associated regions such as the bilateral orbitofrontal cortex and ventral striatum, along with other regions such as the ventromedial prefrontal cortex.
([12])	Binge Eating subgroup, *n* = 30; Control Group, *n* = 29	20.37 ± 1.45; 19.41 ± 1.35	Eating disorder diagnosis scale	Task-based fMRI	Go/no go Task	The binge eating subclinical group relative to controls displayed fewer reward-inhibition undirectional and directional synchronisations (i.e., medial orbitofrontal cortex [mOFC]-superior parietal gyrus [SPG] connectivity, mOFC → SPG excitatory connectivity) during food reward_nogo condition.
([23])	59	25.67 ± 5.11	Eating Disorders Examination (EDE) version 16	Task-based fMRI	Monetary Incentive Delay Task	Mean percent signal change in the right nucleus accumbens was significantly lower in women with BE versus women without BE.
([59])	88	14.5 ± 0.9	Eating Disorder Diagnostic Interview	Task-based fMRI	Food Go/no-go paradigm; Food Receipt Paradigm; Negative Mood Induction Paradigm;	Elevated responsivity of regions implicated in attention and valuation (dorsal anterior cingulate cortex; ventromedial prefrontal cortex) to thin models and lower responsivity of a reward valuation region (caudate) to anticipated milkshake tastes (which correlated with feeling fat) predicted the future onset of binge eating or compensatory behaviours
([70])	Healthy control, *n* = 57; Bulimia nervosa subtypes, *n* = 65; Anorexia nervosa subtypes, *n* = 65	22.63 ± 0.62; 21.70 ± 2.08; 22.21 ± 2.01	Three-Factor Eating Questionnaire	MRI, Task-based fMRI	MID Task; Emotional Face Task; Stop Signal Task	Compared with healthy control participants, eating disorder subgroups showed reduced GMV in the left lateral orbitofrontal cortex and lower cortical thickness in the left rostral middle frontal gyrus and precuneus; eating disorder subgroups exhibited smaller thickness in several left-lateralized regions than HCs, including the left rostral MFG, paracentral lobule, lingual gyrus/precuneus, and left middle temporal gyrus; eating disorder subgroups showed deactivations in the bilateral cerebellum (crus II) and right SFG compared with HCs and lower activations in the visual cortex (left lingual gyrus/right calcarine fissure) during reward anticipation; deactivations in the right middle temporal gyrus and the triangular part of the left IFG were also observed in the BN and AN subgroups, with lower activations or deactivation in other visual areas (right fusiform gyrus and left middle occipital gyrus [MOG]).

^1^ Unless otherwise specified in parentheses, the unit for age is years.

## Data Availability

No new data were created or analysed in this study.

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
