# Peer review of "Early Life Adversity and Disordered Eating: Cognitive and Neural Mechanisms"

_behavsci, 2025, doi:10.3390/bs15121739_

Round 1
Reviewer 1 Report
Comments and Suggestions for Authors
This is an article about an interesting and clinically important topic. However, I have major concerns with this article:
1) It is lacking a methods section. Even a narrative review needs a methods section.
2) Several important studies in the field are not cited. In order to capture those, I recommend that the authors do a systematic review instead of a narrative review with proper search terms, e.g., (over-eating OR binge eating OR obesity) AND (childhood adversity OR childhood trauma) AND (cognition OR neuroimaging OR MRI or connectivity), a clear decision regarding the sources used (e.g., PubMed, ISI Web of Science, PsycInfo), removal of duplicates and an evaluation of the quality of the included studies.
3) The article is completely text only. For readers who find it hard to read just text, a summary table would be helpful. Additionally, a figure that would explain how childhood adversity influences brain circuits and areas, the immune system, the stress system and the intra-cellular signaling regulation would be good.
In summary, even though the topic is important, the selection process for the literature is unclear, important studies have not been mentioned, and the presentation without any table or figure is not appealing.
Author Response
Dear Reviewer 1,
Thank you for your careful review of our manuscript, “Early Life Adversity and Disordered Eating: Cognitive and Neural Mechanisms” (previously titled “Early Life Adversity and Overeating: Cognitive and Neural Mechanisms”). We have revised the manuscript thoroughly in response to your comments. All changes are marked in RED in the revised version. Below, we provide a point-by-point response.
Comment 1: The manuscript lacks a methodology section. Even a narrative review requires one.
Response:
We agree that the lack of a methodological description was a limitation of the earlier version. We have now added a separate Methods section in Part II, detailing the literature search strategy, inclusion and exclusion criteria, and data extraction procedures.
Page 3, line 93:”To identify papers to include in the systematic review, we searched PubMed and Nature database for studies conducted in humans on childhood adversity and/or disordered eating between January 1, 2021, and May 1, 2025. Childhood adversity was indexed using the following search phrases: “early life stress”, “childhood adversity”, “childhood trauma”, “adverse childhood experiences”, “childhood maltreatment”, “emotional abuse”, “emotional neglect”, “physical abuse”, “sexual abuse”, “physical neglect”, “poverty”, and “environmental unpredictability”. Cognitive and neural mechanism was indexed using “cognitive”, “neural”,”fMRI”,”MRI”,”task-based fMRI”. Disordered eating was indexed using “disordered eating”, “overeating”, “emotional eating”, “uncontrolled eating”, “binge eating”. The specific literature search and exclusion criteria are shown in Figure 1.”
Comment 2: A systematic review is recommended instead of a narrative review. The search terms, sources of literature, and quality assessment of included studies should be clearly reported.
Response:
Thank you for this valuable suggestion. We agree that improving the systematicity and transparency of the literature selection process is essential. Although the manuscript retains a narrative-review format, we have substantially strengthened the methodological procedures in line with your advice. The revised version includes a more detailed search strategy, clearer documentation of the screening process, and additional relevant studies identified through the updated search. We also added “Figure 1 PRISMA flow diagram detailing the literature search and selection process for inclusion in the systematic review ” to illustrate the search and selection workflow, which further enhances methodological transparency. See in page 4.
Comment 3: Consider enhancing visualization of the results. A summary table would be highly beneficial, ideally accompanied by figures illustrating how adversity affects neural circuits, the immune system, the stress system, and intracellular signaling pathways.
Response:
Thank you for this concrete suggestion. We agree that visual summaries improve clarity and facilitate information retrieval. Following your recommendation, we have added a summary table —Table 1 Summary for Neuroimaging Studies on Early Life/Childhood Experiences and disordered eating —that organizes key neuroimaging studies included in the review (authors, sample sizes, methods, and main findings). We believe this table substantially improves the structure and readability of the manuscript. See in page 5-15.
We appreciate all three of your comments. In response, we have made substantive revisions to enhance methodological transparency, expand the literature coverage, and improve data visualization. We believe these changes strengthen the rigor and clarity of the manuscript while preserving its central focus.
Thank you once again for your careful review and for the time you devoted to the peer-review process.

Reviewer 2 Report
Comments and Suggestions for Authors
Dear Authors,
Your article “Early Life Adversity and Overeating: Cognitive and Neural Mechanisms” examines how early adverse experiences may contribute to overeating. The manuscript focuses on key neurocognitive aspects, such as the interaction between reward and emotion, inhibitory control, connectivity between the amygdala and prefrontal cortex (PFC), the role of the inferior frontal gyrus (IFG), and alterations in subcortical–cortical (SC–FC) coupling. The review also outlines a fast-track versus slow-track model of brain maturation. The paper takes a narrative approach and does not specify how the literature was identified or selected.
The topic is a good fit for the Behavioral Sciences profile, which focuses on behavioral mechanisms at the intersection of psychology and neurobiology. From the reader's perspective, the article may be valuable as a bridge between the literature on early life adversity and the literature on eating disorders, including obesity. However, the lack of a transparent review methodology and summary tables limits the work's usefulness for researchers and practitioners.
Although the topic is timely and relevant, the manuscript exhibits an evident weakness: the review lacks a defined methodological framework. This gap reduces the scientific strength and transparency of the work. In addition, the discussion relies heavily on publications by the same research group, which raises the risk of citation bias and limits the objectivity of the conclusions. Therefore, I believe it is necessary to introduce a broader, critical review of sources and explicit selection criteria.
In line with current publication standards, even a narrative review should include at least a brief description of its methodology. This typically specifies which databases were searched, the main keywords applied, the time range and language limits, the inclusion and exclusion criteria, the types of studies considered, the process of selecting the literature, and the acknowledged limitations.
The current version of the manuscript is missing a "Methods" section (there is no information on search and selection). This is the paper's main limitation. As a result, it is unclear whether the literature is complete and free from confirmation bias, and the conclusions may be overestimated.
As I have already pointed out, the manuscript is generally well written and has a logical structure. Nevertheless, to improve clarity and scientific quality, several errors and inconsistencies need to be corrected. For example, the abbreviation DMN has been incorrectly expanded as "Default Main Network" instead of "Default Mode Network." The correct form should be updated in the "Abbreviations" section.
There are several linguistic errors (e.g., line 255 – 'This is a neural basis...' should be 'This is a neural basis...') and punctuation errors.
Some references are missing full pages or pagination (e.g., Nature Neuroscience 26(9), "1603-,"; Nature Neuroscience 26(1), "79-").
There are minor inconsistencies in the naming and formatting, for example, in the use of hyphens and italics for variables.
The term "overeating" is used inconsistently and sometimes overlaps with related notions such as emotional eating, binge eating, or uncontrolled eating. These behaviors differ in theory, diagnostic criteria, and the methods used for measurement. The authors use them as if they had the same meaning, which unfortunately blurs key differences and makes the interpretation of the results less straightforward.
The paper would benefit from defining each construct more precisely and citing established definitions or validated questionnaires to support its claims.
Suggestions that are specific and feasible.
- Methods (mandatory):
Add a 'Methods' section containing the following minimum standards:
- Databases (e.g., PubMed, PsycINFO, Web of Science)
- Keywords (ELA/ACE + overeating/binge eating/emotional eating + fMRI/FC/SC)
- Years
- Languages
- Types of studies
- Inclusion/exclusion criteria
- Selection process (two independent reviewers)
- Limitations (narrative vs. systematic) Clearly indicate that this is a narrative review.
- At the beginning of section 3, clarify the operationalisations. Please specify which of the following terms refer to which construct: "overeating", "binge eating", "emotional eating", and "loss of control eating".
- To improve the presentation of the results, it would be beneficial to include a summary table of the most significant studies (cohort, cross-sectional, resting, and task-based fMRI, and SC–FC metrics). The table should have the following columns: population, age, ELA measure, eating construct, neural measure, key findings, and limitations.
Adding a table or figure that shows the separation of results according to reward processing phases (anticipation, consumption, and decision) would also enhance the value of the work. A graphical version would make it easier for the reader to interpret, which is already present.
- Include a subsection on moderators/confounders: gender, age, SES, comorbidities (ADHD, depression/medication), sleep, medication, BMI; indicate where data are contradictory or incomplete.
- Expand the literature with works outside the circle of authors; add studies that weaken or do not confirm the hypothesis (important for balance).
- In the discussion, correlations should be clearly separated from causality, and then the need for longitudinal studies measuring ELA → brain development trajectories → eating behavior trajectories should be emphasized. Finally, attention should be drawn to the gaps (childhood and adulthood, as well as the transition from adolescence to early adulthood).
After introducing a methods section, expanding and balancing the literature, adding tables/figures, and undergoing editorial revision, the manuscript could become a valuable, well-cited item in Behavioral Sciences.
Kind regards
Reviewer
Author Response
Dear Editor and Reviewer 2,
We encountered a technical issue while submitting our response, as the system only allowed a limited portion of the text to be entered (possibly due to a word limit). The full response is provided in the attached file. We apologize for any inconvenience this may have caused.

Round 2
Reviewer 1 Report
Comments and Suggestions for Authors
The authors have responded well to the questions and comments raised.
Author Response
We sincerely thank you for your recognition of our work and for the insightful and constructive suggestions you provided during the revision process. We also greatly appreciate the time and effort you devoted to reviewing our manuscript. We sincerely wish you all the best in your future endeavors.
Reviewer 2 Report
Comments and Suggestions for Authors
Dear Authors,
In its current form, the manuscript "Early Life Adversity and Overeating: Cognitive and Neural Mechanisms" has been significantly improved. The Authors have clearly addressed most of the previous comments, improving the structure of the review, filling in the missing elements, and increasing the clarity of the article. Thanks to a more straightforward presentation of the theoretical model and the reorganization of the methodology section, with the addition of a description of the search procedure and the PRISMA flow diagram, the manuscript has become much more transparent. The whole is now more in line with the journal's standards and can make a valuable contribution to the discussion on the neural and cognitive mechanisms linking early trauma to eating disorders.
However, the manuscript still requires a series of linguistic and editorial revisions, as well as a few minor substantive additions to increase its value and professional reception.
Areas that still need refinement:
a. The text contains some minor errors, including typos, incorrect word divisions, missing words, and the occasional non-English character.
These include, among others:
Line 264 - is 'This is a neural basis...' should be 'This is a neural basis...'
Lines 265, 285, and 289 - is 'disordered', 'symptoms', 'likelihood' should be 'disordered', 'symptoms', 'likelihood'.
Full-width comma: 'However, ...'
Line 186: the incorrect form "iu et al., 2020".
A thorough correction by someone proficient in English is recommended.
I recommend that the Authors clarify the description of the literature search to improve methodological transparency.
The methods section should explain what is meant by the phrase "searched PubMed and nature", since this suggests the name of a journal rather than a database.
It is worth adding a brief summary of the inclusion and exclusion criteria in the text, rather than limiting yourself to a reference to the figure.
Terminology: 'Disordered eating' vs. 'Overeating.'
The term' Overeating Behaviours' appears in several places. It would be helpful to briefly explain the relationship between these concepts (e.g., whether 'overeating' is a subtype of 'disordered eating' or whether it is an independent area that has been included due to the scope of the study).
It would also be advisable to include a sentence clearly referring to gaps in the literature concerning:
- younger children;
- adults,
- the transition period between adolescence and early adulthood. The authors mention this observation in their response letter, but they do not state it explicitly in the manuscript.
5. It is advisable to check that abbreviations (e.g., ELA/ELS, RSFC, fMRI) are consistent between the table, text, and list of abbreviations.
The manuscript is now solid and significantly refined in terms of content.
It requires only minor corrections. These are mainly linguistic and editorial. There are also a few clarifications needed. These are in the methods and discussion sections. Once the authors implement these revisions, the paper will be ready for publication in Behavioral Sciences.
Best od luck,
Reviewer
Comments on the Quality of English LanguageThe English is clear and of good quality, with only minor issues related to phrasing and stylistic consistency. A light editorial revision would further polish the manuscript.
Author Response
Response to Reviewer 2
Dear Reviewer 2:
Thank you for your thoughtful and constructive evaluation of our manuscript. We have carefully addressed each of your comments in detail below and implemented the corresponding revisions in the manuscript. Substantive changes have been highlighted in RED for your convenience. Minor editorial adjustments to wording and formatting were also made but, due to their limited scope, are not individually marked.
We sincerely appreciate the time and effort you invested in reviewing our work.
- Linguistic & Editorial Issues
(1) The text contains some minor errors, including typos, incorrect word divisions, missing words, and the occasional non-English character.
These include, among others:
Line 264 - is 'This is a neural basis...' should be 'This is a neural basis...'
Lines 265, 285, and 289 - is 'disordered', 'symptoms', 'likelihood' should be 'disordered', 'symptoms', 'likelihood'.
Full-width comma: 'However, ...'
Line 186: the incorrect form "iu et al., 2020".
A thorough correction by someone proficient in English is recommended.
It is advisable to check that abbreviations (e.g., ELA/ELS, RSFC, fMRI) are consistent between the table, text, and list of abbreviations.
Response to comments:
Thank you for carefully identifying these typographical and formatting issues. We have corrected all items you noted, including the errors on lines 186, 264, 265, 285, and 289, the full-width comma, and the incorrect citation format. We also reviewed and ensured the consistency of abbreviations across the tables, main text, and the list of abbreviations.
To further ensure accuracy and clarity, we engaged a professional English editing service to perform a thorough language check of the entire manuscript. Minor editorial adjustments to wording and formatting were also made; due to their limited scope, these changes are not individually highlighted in red in the main manuscript.
- Methodological transparency issues
(1) I recommend that the Authors clarify the description of the literature search to improve methodological transparency.
The methods section should explain what is meant by the phrase "searched PubMed and nature", since this suggests the name of a journal rather than a database.
Response to comments:
Thank you for highlighting this issue. We agree that listing “Nature” alongside PubMed could be misleading. We have revised the Methods section to clarify that “Nature” refers to targeted searches within the Nature family of journals and citation tracking, rather than a database. The manuscript now explicitly specifies the databases used and provides a clearer description of our literature search strategy to enhance methodological transparency.
Page 3, line 90 : The full screening workflow is presented in Figure 1. To identify papers to include in the systematic review, we searched PubMed database and Nature Affiliated journals for studies conducted in humans on childhood adversity and disordered eating between January 1, 2015, and May 1, 2025.
(2)It is worth adding a brief summary of the inclusion and exclusion criteria in the text, rather than limiting yourself to a reference to the figure.
Response to comments:
Thank you for this helpful suggestion. We agree that including a concise summary of the inclusion and exclusion criteria in the main text enhances clarity. Accordingly, we have added a brief description of these criteria in the Methods section, complementing the detailed information presented in the figure.
Page 3, line 99:We included studies that: (1) examined early life adversity (ELA), or subclinical disordered eating symptoms/disordered eating; (2) used neuroimaging methods (structural MRI, resting-state fMRI, or task-based fMRI). We excluded studies that: (1) did not involve neuroimaging, such as genetic research or behavioural research; or (2) focused on clinically diagnosed populations, including eating disorders or other psychiatric or medical conditions.
The initial search yielded 4,978 records (PubMed: 3,514; Nature and affiliated journals: 1,464). After title and abstract screening, 3,924 records were removed for being clearly unrelated to the topic. An additional 571 news items, commentaries, or narrative reviews were excluded. We then removed 3,353 records that did not involve human neuroimaging. A total of 1,054 articles underwent full-text assessment, during which 966 were excluded because they involved clinical populations or did not meet other inclusion criteria. After removing 41 duplicates, 47 studies met all criteria and were included in the final analysis.
- TerminologyClarification: 'Disordered eating' vs. 'Overeating.'
The term' Overeating Behaviours' appears in several places. It would be helpful to briefly explain the relationship between these concepts (e.g., whether 'overeating' is a subtype of 'disordered eating' or whether it is an independent area that has been included due to the scope of the study).
Response to comments:
Thank you for raising this point. We clarified at the beginning of the manuscript that overeating is considered one form of disordered eating in the context of this review. We have revised the terminology throughout the paper to ensure consistent use of these concepts. We appreciate your careful attention to this issue. Corresponding clarifications have been added to the Introduction and Methods sections to ensure terminological consistency.
Page 1, line 23: Disordered eating refers to a broad range of maladaptive, psychologically driven eating behaviours that significantly impair physical health and psychosocial functioning (Chen & Jackson, 2008).
- Discussion of Limitations
It would also be advisable to include a sentence clearly referring to gaps in the literature concerning:
younger children;
adults,
the transition period between adolescence and early adulthood. The authors mention this observation in their response letter, but they do not state it explicitly in the manuscript.
Response to comments:
Thank you for drawing our attention to this important point. We have now included an explicit statement in the Limitations section acknowledging the limited evidence available for younger children, adults, and individuals transitioning from adolescence to early adulthood. In addition, we have made minor revisions to both the Methods and Limitations sections to ensure that these gaps in the current literature are clearly and accurately reflected in the manuscript..
Page 4, line 113: (methods section) Among the 47 studies included, the age composition of samples differed substantially. Based on the mean age reported in each study, 7 examined early childhood groups (0–6 years), 20 involved school-age or adolescent samples, and 20 focused on adult participants. Notably, because early-life adversity is generally defined as exposure occurring before age 16.
Page 5, line 337:(limitations section) Finally, most studies included in this review focused on early childhood, adolescence, and young adulthood. This concentration in younger samples should be noted, as it limits the generalisability of the findings to later developmental periods.